# A metrologically traceable protocol for the quantification of trace metals in different types of microplastic

Lars Hildebrandt[1,2], Marcus von der Au[3,4], Tristan Zimmermann[1], Anna Reese[1,2], Jannis Ludwig[5], Daniel Pröfrock[1] *

1 Marine Bioanalytical Chemistry, Institute of Coastal Research, Helmholtz-Zentrum Geesthacht, Geesthacht, Germany, 2 Department of Chemistry, Inorganic and Applied Chemistry, Universität Hamburg, Hamburg, Germany, 3 Department G2—Aquatic Chemistry, Federal Institute of Hydrology, Koblenz, Germany, 4 Division 1.1—Inorganic Trace Analysis, Federal Institute for Materials Research and Testing, Berlin, Germany, 5 Department of Chemistry, Institute of Organic Chemistry, University of Kiel, Kiel, Germany

* daniel.proefrock@hzg.de

## Abstract

The presence of microplastic (MP) particles in aquatic environments raised concern about possible enrichment of organic and inorganic pollutants due to their specific surface and chemical properties. In particular the role of metals within this context is still poorly understood. Therefore, the aim of this work was to develop a fully validated acid digestion protocol for metal analysis in different polymers, which is a prerequisite to study such interactions. The proposed digestion protocol was validated using six different certified reference materials in the microplastic size range consisting of polyethylene, polypropylene, acrylonitrile butadiene styrene and polyvinyl chloride. As ICP-MS/MS enabled time-efficient, sensitive and robust analysis of 56 metals in one measurement, the method was suitable to provide mass fractions for a multitude of other elements beside the certified ones (As, Cd, Cr, Hg, Pb, Sb, Sn and Zn). Three different microwaves, different acid mixtures as well as different temperatures in combination with different hold times were tested for optimization purposes. With the exception of Cr in acrylonitrile butadiene styrene, recovery rates obtained using the optimized protocol for all six certified reference materials fell within a range from 95.9% ± 2.7% to 112% ± 7%. Subsequent optimization further enhanced both precision and recoveries ranging from 103% ± 5% to 107 ± 4% ($U$; $k = 2$ ($n = 3$)) for all certified metals (incl. Cr) in acrylonitrile butadiene styrene. The results clearly show the analytical challenges that come along with metal analysis in chemically resistant plastics. Addressing specific analysis tools for different sorption scenarios and processes as well as the underlying kinetics was beyond this study's scope. However, the future application of the two recommended thoroughly validated total acid digestion protocols as a first step in the direction of harmonization of metal analysis in/on MP will enhance the significance and comparability of the generated data. It will contribute to a better understanding of the role of MP as vector for trace metals in the environment.

**Data Availability Statement:** All relevant data are within the manuscript and its Supporting Information files.

**Funding:** Anna Reese was funded by the BSH through the project OffChEm (BSH contract code: 10036781, HZG contract code: 17/2017).

**Competing interests:** The authors have declared that no competing interests exist.

## 1. Introduction

Anthropogenic litter, especially highly persistent plastic litter, has become a global problem [1–3]. The hypothesis that microplastic (MP) may be a vector for potentially harmful chemicals ("Trojan horse effect") has gained "paradigm status" among scientists focusing on the occurrence and effects of MP [4–7]. Numerous studies have examined in particular the sorption of hydrophobic organic chemicals, such as polycyclic aromatic hydrocarbons, dioxins, phthalates and polychlorinated biphenyls to MP [8–11] and their further transfer to organisms via MP [12–15].

In general, the toxicity of heavy metals such as As, Cd, Cr, Hg, Pb, Sb, Sn present in the marine environment is well studied and documented [16–18]. Nevertheless, in contrast to the MP-mediated transport of organic contaminants, the analogous transport of metals and the related adsorption and desorption processes remain poorly understood. Two scenarios are currently being considered in this context.

Firstly, inorganic and organic metal compounds are introduced to plastics as additives during manufacturing to adjust their specific properties, i.e. as heat stabilizers, fillers, pigments, biocides, flame retardants, slip agents [19–27], or as polycondensation catalyst in case of industrial PET production ($Sb_2O_3$) [28].

Secondly, a comparably new finding is that metal ions sorb to MP in water bodies [29, 30]. Several studies have focused on the mass fractions of selected metals associated with MP particles that were collected in the environment (Table 1). The US Environmental Protection Agency has published a white paper in 2016 that addresses the scientifically observed sorption of metals to plastics in aquatic systems [31]. The authors of this white paper call for more research to elucidate the role of sorption and transfer of toxic heavy metals via MP.

**Table 1. Overview of the existing publications on the metal content of MP collected in the (aquatic) environment.**

| Author | Metals | MP size (d) | Sorbate | Reagents for digestion | Certified reference material | Digestion recoveries | Analytical technique |
|---|---|---|---|---|---|---|---|
| Ashton et al. (2010) [32] | Ag, Al, Cd, Co, Cr, Cu, Fe, Mn, Mo, Pb, Sb, Sn, U, Zn | 3–5 mm | PE pellets from beaches in England | 2 M HCl and 3 M HNO₃ (3:1) | LKSD 4 (sediment) | 70% - 80% for Al, Co, Fe and Mo | ICP-MS, ICP-OES |
| Holmes et al. (2012) [33] | Cd, Cu, Co, Cr, Ni, Pb, Zn | > 1 mm | PE and PP pellets from beaches in England | 20% aqua regia | - | - | ICP-MS, ICP-OES |
| Turner and Holmes (2015) [34] | Ag, Cd, Co, Cr, Cu, Hg, Ni, Pb, Zn | ~ 4 mm | PE and PP pellets from beaches in England | 20% aqua regia | - | - | ICP-MS (Collision cell) |
| Vedolin et al. (2017) [35] | Al, Cr, Cu, Fe, Mn, Sn, Ti, Zn | n.i. | PE and PP pellets from beaches in Brazil | HNO₃, HCl, H₂O₂ (all conc.) | SS-1 EnvironMAT SPC Science -Contaminated Soil | - | ICP-OES |
| Wang et al. (2017) [36] | Cd, Cu, Ni, Pb, Ti, Zn | n.i. | MP extracted from littoral sediments of a river in China (density separation with NaCl solution) | H₂O₂, HNO₃, H₂SO₄ (all conc.) | GBW1004, carrot GBW10044, rice | 90%– 113% | ICP-MS |
| Munier and Bendell (2018) [37] | Cd, Cu, Pb, Zn | Micro- and macroplastic | Items from beaches in Canada | 10% HNO₃ | - | - | AAS |
| Wijesekar et al. (2018) * [38] | Cd, Cu, Ni, Pb, Sb, Zn | < 50 μm– 1000 μm | Microbeads from biosolids collected in Australia (density separation with BaCl₂ solution) | aqua regia | NIST 1643e and NIST 1643 (water standards) | - | ICP-MS |

* Study investigating MP extracted from biosolids.

However, none of the studies listed in Table 1 provides recoveries for the applied digestion or extraction protocol based on usage of a matrix-matched certified reference material (CRM). Even though there are different plastic CRMs for metal analysis available, some studies have used no CRM, whereas others employed unsuited, non-matrix matched CRMs (e.g. water, sediments, soil, sewage sludge, rice or carrot CRMs). Application of such non- or poorly (according to international metrology standards) validated procedures leads to generation of inaccurate, non-traceable and incomparable data. Therefore, in analytical chemistry, using matrix-matched CRMs is indispensable for the generation of comparable and metrologically traceable data as well as the calculation of uncertainty budgets according to the "Guide to the expression of uncertainty in measurement" (GUM (JCGM 100:2008)) [39]. The formal definition of "uncertainty of measurement" would be: "parameter, associated with the result of a measurement, that characterizes the dispersion of the values that could reasonably be attributed to the measurand" [39] (measurand in this context may be replaced with concentration for most areas of chemical analysis).

Expanded uncertainties take into account all major potential error contributions (e.g. measurement precision, reproducibility, inhomogeneity of the sample, blank contribution) (Fig 2) and a coverage factor (in the case of assumed normal distribution using ± two combined uncertainties refers to a 95.4% confidence interval). Therefore, uncertainties will not only give a measure of the quality of a result enabling the user to assess the reliability of analytical data, they also facilitate identification of the significant sources of uncertainty in a measurement procedure. Only if there is no overlap of the referring confidence intervals of two means, effects are significant based on a predefined significance level (α). For meaningful assessment of the data on the interactions between metals and MP but also for data on the general abundance of MP particles and fibers [40], thorough method validation and harmonized protocols are needed, including reference materials, inter-laboratory comparison tests and sound applications of existing metrological-analytical concepts.

Weak acidic extraction/leaching protocols [32, 37], for instance, bear different degrees of selectivity towards different metals and metal species. Additionally, the degree of desorption (achieved by leaching) can vary between different polymer types (depending on the chemical structure of the polymeric chain). Maybe even more importantly, a meaningful assessment of the sorption and desorption behavior cannot be conducted without considering a variety of physical parameters, e.g. permeability, diffusion coefficients, solubility and polarity [41]. Müller *et al.* (2018) have demonstrated that sorption (and herewith also desorption) of chemicals to MP is highly influenced by polymer-specific parameters such as glass transition temperature and crystallization content [42].

To overcome resulting selectivity differences, it is advisable to put future studies focusing on the role of MP as a vector for metal contaminants either on the basis of a complete microwave-assisted acid digestion (MWAD) protocol [43] or the application of techniques such as laser ablation inductively coupled plasma mass spectrometry (LA-ICP-MS) for direct surface analysis (in this case a polymer-type-specific matrix matched calibration strategy would be also required for accurate trace metal quantification).

There is already scientific evidence (from polymer testing studies) that shows how challenging the accurate quantification of metals in CRMs of different polymer matrices is [44–48]. Dependent on the polymer type, the metal species and the applied digestion conditions, the recoveries can highly vary from a few percent to a quantitative recovery [47, 49].

To overcome these difficulties, this study presents the development of a new, validated MWAD approach for metal analysis in the most important polymeric matrices on the market (in terms of production volume) using five CRMs from different international and national metrology and research institutes, and one certified quality control standard from a chemical

company. The polymer types polyethylene (PE), polypropylene (PP) and polyvinyl chloride (PVC), which were investigated in this study, account for > 60% of the global plastic production volume [50] and a high share of the MP particles typically detected in aquatic environments [51–53]. Despite a market share < 3% [50], acrylonitrile butadiene styrene (ABS) was also investigated as a representative of styrene copolymers, since > 40% of plastics in electrotechnical waste are assigned to it [54, 55]. Electrotechnical waste can contain remarkable mass fractions of heavy metals [54, 55] and is often practically unrecyclable.

The aim of this study is to provide a thoroughly validated polymer-specific MWAD protocol for metal analysis in MP for a better understanding of the interactions between MP sampled in different environments and metal contaminants. The protocol provides a basis for the generation of comparable data, which is a primary prerequisite to study the large scale role of interactions between metal contaminants and MP in the environment.

## 2. Experimental

All experiments based on the MARS 6 and the Discover SP-D 35 (CEM Corp., Kamp Lintfort, Germany) microwaves was carried out at the Helmholtz-Zentrum Geesthacht. Four digestion batches of the CRM BAM-H010 using the turboWAVE (MLS GmbH, Leutkirch, Germany) and the subsequent multi-elemental analysis were conducted at the Federal Institute of Hydrology. All procedures were performed under clean room conditions. The three microwave systems compared in this study differ in the general construction, but the main practical differences refer to the number of vessels that can be processed at a time, the vessel sizes (section 2.2) and the pressure as well as temperature regulation. Briefly summarized, the MARS 6 and the Discover SP-D 35 (external IR temperature control; pressure vessels) used in this study enable digestion at temperatures up to 230˚C and observed pressures up to 24–28 bar, whereas the turboWAVE bears a significantly higher maximum temperature of 300˚C and also a significantly higher maximum pressure of 200 bar. In the tuboWAVE, Temperature and pressure are both regulated and controlled in a single reaction chamber filled with inert gas. In contrast to the MARS 6 and the turboWAVE microwave, that feature simultaneous processing of a batch of digestions (40 and 15 vessels), the Discover SP-D 35 (in conjunction with an Explorer autosampler) irradiates the vessels automatically one after another enabling variation of digestion parameters for method development (different conditions for every vessel possible). Please note that this comparison is not meant to be exhaustive, since there are a lot of different vessel types (e.g. for different maximum pressures and temperatures), add-ons features (e.g. for pressure and temperature control) and also other microwave systems available on the market.

### 2.1 Reference materials, reagents and solutions

**Polymer certified reference materials.** The two PE CRMs ERM®-EC680m and ERM®-EC681m (elements, low and high level) were purchased from the Joint Research Centre of the European Commission (JRC, Ispra, Italy), a PVC (NMIJ CRM 8123-a) and a PP (NMIJ CRM 8133-a) CRM from the National Metrology Institute of Japan (NMIJ, Tsukuba, Japan), an ABS CRM (BAM-H010) from the Federal Institute for Materials Research and Testing (BAM, Berlin, Germany) and another PP CRM (Lead in Plastic–QC (trade name)) from Sigma Aldrich (Wyoming, USA).

The chemical structures of the four polymer types are shown in Fig 1. Herewith, the six CRMs refer to four different polymer types and provide certified mass fractions for one to eight metals, while covering a concentration range for selected analytes of five orders of magnitude (Table 2).

The particle sizes of these polymer CRMs were determined using a PALM MicroBeam Microscope (Carl Zeiss AG, Oberkochen, Germany). Lead in Plastic—QC (Sigma Aldrich,

**Fig 1. Chemical structures of the four corresponding polymer types that were covered by the six CRMs.**

Wyoming, USA) can be assigned to the small MP size range ($<$ 500 μm), whereas the other five of the six used CRMs fall within the large MP size range (500 μm—5 mm) [56, 57].

**Procedures conducted at the Helmholtz-Zentrum Geesthacht.** Laboratory work was performed in a class 10,000 clean room inside a class 100 clean bench. Type I reagent-grade water (18.2 MΩ cm) was obtained from a Milli-Q Integral water purification system (Merck-Millipore, Darmstadt, Germany) equipped with a Q-Pod Element system. P.a. grade nitric acid ($HNO_3$) (65% *w/w*, Merck-Millipore) and hydrochloric acid (HCl) (30% *w/w*, Merck-Millipore) were further purified by double sub-boiling in PFA stills (Savillex, Eden Prairie, USA). Tetrafluoroboric acid ($HBF_4$) (38% *w/w*, Chem-Lab, Zedelgem, Belgium) and hydrogen peroxide ($H_2O_2$) (30% *w/w*, ultrapure, Merck-Millipore) were used for sample digestion without any further purification. Polyethylene (PE) flasks, tubes and pipette tips (VWR International, Radnor, USA), as well as perfluoroalkoxy (PFA) screw cap vials (Savillex, Eden Prairie, USA) were pre-cleaned in a two-stage washing procedure using diluted $HNO_3$ (10% *w/w* and 1% *w/w* respectively). Microwave vessels were cleaned by running the respective MWAD

**Table 2. Overview of used polymer CRMs.**

| Name of CRM | Polymer type | Shape | Size [μm] * | Certified metals | Mass fraction range [mg kg$^{-1}$] ** |
|---|---|---|---|---|---|
| **Lead in Plastic—QC** | Polypropylene | Powder | 90 ± 80 ($d_{max}$) | Pb | 376.0 ± 18.9 |
| **ERM®- EC680m** | Polyethylene | Pellets | (2500 ± 100) × (2960 ± 30) ($h \times w$) | As, Cd, Cr, Hg, Pb, Sb, Sn, Zn | 2.56 ± 0.16–194 ± 12 |
| **ERM®-EC681m** | Polyethylene | Pellets | (3760 ± 130) × (2640 ± 120) ($h \times w$) | As, Cd, Cr, Hg, Pb, Sb, Sn, Zn | 7.0 ± 1.2–1170 ± 40 |
| **NMIJ CRM 8123-a** | Polyvinyl chloride | Pellets | (3220 ± 70) × (1700 ± 100) ($h \times w$) | Cd, Cr, Hg, Pb | 95.62 ± 1.39–965.5 ± 6.6 |
| **NMIJ CRM 8133-a** | Polypropylene | Spherules | 4320 ± 160 ($d$) | Cd, Cr, Hg, Pb | 94.26 ± 1.39–949.2 ± 7.5 |
| **BAM-H010** | Acrylonitrile butadiene styrene | Pellets | (3130 ± 70) × (2770 ± 60) ($h \times w$) | Cd, Cr, Pb; Information value for Hg | 93 ± 5–479 ± 17 |

* 1 SD ($n_{2-6}$ = 3, $n_1$ = 50)

** $U_{certified}$ ($k$ = 2).

program two times solely with 4 mL $HNO_3$ and 1 mL HCl (without CRM). Subsequently, the vessels were washed 3-times with Milli-Q water.

**Procedures conducted at the Federal Institute of Hydrology.** The deionized water (18.2 MΩ cm) used was obtained from a Milli-Q water purification system (Merck-Millipore). For sample digestion, the suprapur® nitric acid and suprapur® hydrochloric acid (65% *w/w* and 30% *w/w*, respectively, both Merck-Millipore) used were further purified by sub-boiling in PFA stills (Savillex). Hydrogen peroxide (30% *w/w*, ultrapure, Merck-Millipore) and tetrafluoroboric acid (48% *w/w*, Sigma Aldrich, Saint Louise, USA) were used without further purification. The PE tubes and the pipette tips (VWR International) were pre-cleaned in 1.3% *w/w* $HNO_3$.

## 2.2 Digestion of certified reference materials

About 100 mg of the reference materials ERM®-EC680m, NMIJ CRM 8123-a, NMIJ CRM 8133-a, BAM-H010 and Lead in Plastic—QC, and about 60 mg of ERM®-EC681m were weighed into pre-cleaned 35 mL quartz pressure vials (Discover SP-D 35) or 55 mL MARSXpress TFM® (trade name; cross-linked $[(CF_2)_4\text{-}CF(\text{-}O\text{-}CF_2\text{-}CF_2\text{-}CF_3)\text{-}(CF_2)_5]_n$) bombs (MARS 6) (both CEM Corp.), respectively. The respective amount of $HNO_3$, HCl, $H_2O_2$ and $HBF_4$ (section 2.1) was added to the microwave vessels containing the CRM. The samples were digested for 15 min up to 80 min at 210°C to 230°C using both the Discover SP-D 35 and the MARS 6 microwave systems. Temperatures of 260°C and 300°C were set for the digestions using the turboWAVE (MLS GmbH) in conjunction with 24 mL TFM® vessels (17 min ramp and 30 min hold time). After digestion, the solution was transferred quantitatively to a 50 mL pre-cleaned DigiTUBE (SCP Science, Quebec, Canada) and diluted to a final volume of 50 mL with Milli-Q water.

## 2.3 Instrumentation, measurement routines and data processing

**2.3.1 Multi-elemental analysis.** Multi-elemental analyses of the samples were performed using an inductively coupled plasma—tandem mass spectrometry (ICP-MS/MS) instrument (Agilent 8800, Agilent Technologies, Tokyo, Japan) either coupled to an ESI SC-4 DX FAST autosampler (Elemental Scientific, Omaha, USA) at the Helmholtz-Zentrum Geesthacht or to an SPS 4 autosampler (Agilent Technologies) at the Federal Institute of Hydrology. Both instruments were optimized in a daily routine using a tuning solution, containing Li, Co, Y, Ce, Tl or Be, In, Ce and U to maintain a reliable day-to-day-performance. Rh and Ir were used as internal normalization standards (Merck-Millipore).

General instrumental settings for the multi-elemental measurements are described in S1 Table in S1 File. Best suitable detection modes ([no gas], [He], $[O_2]$ or $[H_2]$) and isotopes were chosen according to recoveries for the in-house quality control multi-element standard solution (Inorganic Ventures), that was rigorously measured at least five times during each measurement batch (S2 Table in S1 File).

**2.3.2 Data processing.** Multi-elemental data were processed using Mass Hunter version 4.4 (Agilent Technologies, Tokyo, Japan) and a custom written MS Excel© spreadsheet. An outlier evaluation after Dixon's Q Test [58] was utilized. A Q value of 0.559 (*n* = 6) was used for outlier evaluation (90% confidence interval).

External linear calibration was applied for quantification. Limits of detection (*LOD*) and limits of quantification (*LOQ*) of the method were calculated in accordance with MacDougall *et al.* (1980) [59] from procedural blanks (*n* = 3) (S2 Table in S1 File). Combined uncertainties ($u_c$) were calculated for representative samples using a *Kragten* spreadsheet approach and are reported as expanded uncertainties (*U*; *k* = 2) (S3 and S4 Tables in S1 File) [60, 61]. The calculations included the error of weight, error of the volume, as well as instrument (measurement precision) and sample replicates (repeatability) (Fig 2).

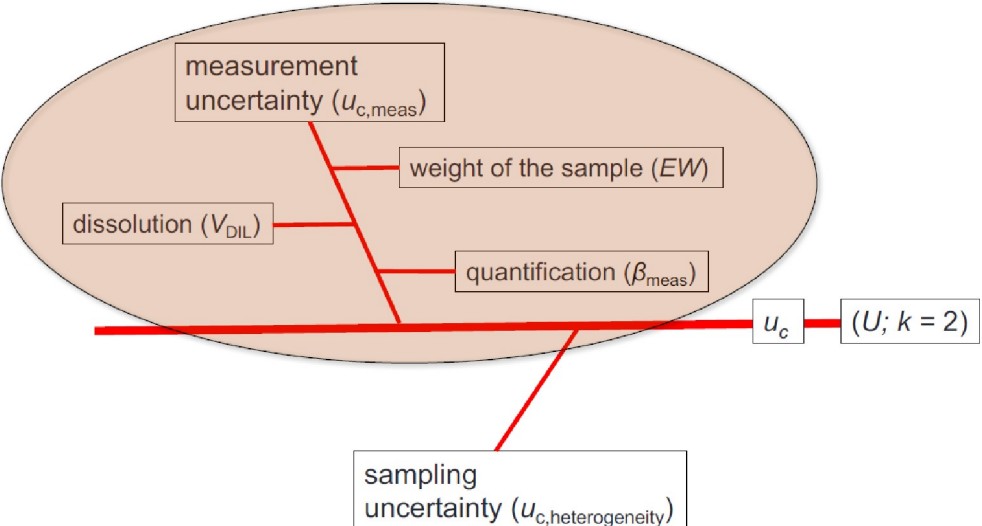

**Fig 2. Fishbone diagram showing the different contributors to the combined uncertainty ($u_c$) of ICP-MS/MS measurements ($U$: expanded uncertainty; $k$: coverage factor) [62].** All error contributing to the overall uncertainty must be considered.

Details about the calculation of combined uncertainties for the certified elements can be found in S3 and S4 Tables in S1 File. The significant number of digits of mass fractions are given according to the Guide to the expression of Uncertainty in measurement (JCGM 100:2008) [39] and EURACHEM guidelines [63], whereby the uncertainty determines the significant number of digits to be presented with the value. To evaluate the performance of the analytical procedure, *zeta* scores (Eq 1) were calculated according to ISO/IEC Guide 43–1:1997 § A.2.1.4 and ISO/DIS 13528 2002 [64, 65]. |*zeta*| scores below 2 indicate satisfactory results.

$$zeta = \frac{x_{lab} - x_{ref}}{\sqrt{U_{lab}^2 + U_{ref}^2}} \tag{1}$$

Eq 1: Calculation of the *zeta* score as an important performance indicator.

**2.3.3 Nuclear magnetic resonance spectroscopy.** $^1$H NMR spectroscopy was applied to characterize the precipitate that occurred during the digestion of BAM-H010. It was carried out on a Bruker Avance 200 at 200 MHz in DMSO-$d_6$ and tetramethyl silane (TMS) was added as internal reference (TMS: $\delta$ = 0.00 ppm). Prior to spectroscopic analysis, decomposed ABS was recrystallized from water/ethanol (4:1) after hot filtration.

## 3. Results and discussion

### 3.1 Method optimization using Discover SP-D 35 and MARS 6 microwaves

Good results in terms of trueness and precision for all six CRMs were achieved by means of a combination of 4 mL HNO$_3$ and 1 mL HCl, and digestion at a temperature of 230°C using the MARS 6 Microwave (20 min ramp and 60 min hold time) (Table 3).

The use of sulfuric acid (H$_2$SO$_4$) was omitted in order to avoid non-spectral interferences [66] and reduced recoveries for Pb, due to the formation of insoluble sulfates [47]. In General, the addition of H$_2$O$_2$ and HBF$_4$ to the mixture of HNO$_3$ and HCl did not lead to better

**Table 3. Mass fractions for the six digested certified reference materials using two different microwave systems and acid mixtures (5 mL HNO$_3$ for Discover SP-D 35 and 4 mL HNO$_3$ + 1 mL HCl for MARS 6).**

Mass fraction ($w$ [mg kg$^{-1}$] ($U$; $k = 2$ ($n = 6$)))

| CRM | ERM®-E680m | | | ERM®-E681m | | | BAM-H010 | | | NMIJ CRM 8123-a | | | NMIJ CRM 8133-a | | | Lead in Plastic-QC*** | |
|---|---|---|---|---|---|---|---|---|---|---|---|---|---|---|---|---|---|
| Metal | Cert. value | Discover SP-D 35 | MARS 6 | Cert. value | Discover SP-D 35 | MARS 6 | Cert. value | Discover SP-D 35 | MARS 6 | Cert. value | Discover SP-D 35 | MARS 6 | Cert. value | Discover SP-D 35 | MARS 6 | Cert. value | MARS 6 |
| As | 4.7 ± 0.4 | 5.03 ± 0.29 | 4.7 ± 0.7 | 17.0 ± 1.2 | 19.5 ± 2.0 | 16.8 ± 1.7 | - | < LOD | < LOD | - | < LOD | < LOD | - | < LOD | < LOD | - | 0.14 ± 0.09 (< LOQ) |
| Cd | 20.8 ± 0.9 | 24.9 ± 1.8 | 21.7 ± 1.7 | 146 ± 5 | 187 ± 19 | 151 ± 9 | 93 ± 5 | 150 ± 13 | 103 ± 5 | 95.62 ± 1.39 | 110.0 ± 2.8 | 98 ± 6 | 94.26 ± 1.39 | 107 ± 3 | 96 ± 6 | - | 0.043 ± 0.027 |
| Cr | 9.6 ± 0.5 | 12.9 ± 0.8 | 9.6 ± 0.8 | 45.1 ± 1.9 | 55 ± 6 | 44.6 ± 2.0 | 470 ± 36 | 430 ± 250 | 1.4 ± 0.4 | 949.0 ± 9.7 | 980 ± 40 | 950 ± 90 | 895.2 ± 9.6 | 1050 ± 60 | 930 ± 50 | - | 22 ± 7 |
| Hg* | 2.56 ± 0.16 | - | 2.7 ± 0.4 | 9.9 ± 0.8 | - | 10.1 ± 1.0 | 415 ± 27 ** | - | 404 ± 20 ** | 937.0 ± 19.4 | - | 1020 ± 50 | 941.5 ± 19.6 | - | 1050 ± 60 | - | < LOD |
| Pb | 11.3 ± 0.4 | 14.0 ± 0.9 | 11.6 ± 1.2 | 69.7 ± 2.5 | 91 ± 9 | 71 ± 5 | 479 ± 17 | 787 ± 76 | 520 ± 30 | 965.5 ± 6.6 | 1110 ± 30 | 970 ± 50 | 949.2 ± 7.5 | 1080 ± 30 | 1000 ± 80 | 376.0 ± 18.9 | 360 ± 10 |
| Sb | 9.6 ± 0.7 | 9.8 ± 3.4 | 10.0 ± 0.9 | 86 ± 7 | 71 ± 7 | 90 ± 5 | - | < LOD | < LOD | - | < LOD | < LOD | - | < LOD | < LOD | - | 0.06 ± 0.04 |
| Sn | 20.7 ± 1.6 | 5.4 ± 1.4 | 21.5 ± 2.0 | 99 ± 6 | 23 ± 7 | 102 ± 7 | - | < LOD | < LOD | - | < LOD | < LOD | - | < LOD | < LOD | - | 4.2 ± 1.2 |
| Zn | 194 ± 12 | 231 ± 14 | 205 ± 16 | 1170 ± 40 | 1420 ± 130 | 1210 ± 80 | - | < LOD | < LOD | - | 575 ± 17 | 510 ± 150 | - | 121 ± 4 | 112 ± 11 | - | 16 ± 4 |
| \|zeta\| range | - | 0.06–7 | 0.04–0.6 | - | 1.1–9 | 0.08–0.5 | - | 0.17–4 | 0.3–13 | - | 1.7–4 | 0.0019–1.5 | - | 2–5 | 0.3–1.7 | - | 0.7 |

* Hg was not measured for the Discover SP-D 35

** Information value for BAM-H010

*** Not digested using Discover SP-D 35.

recoveries (Fig 3) but—in case of $H_2O_2$—an increase of pressure in the microwave vessels and higher losses of acid mixture during the digestion were observed. Moreover, an automatic release of the vessel pressure above approximately 24 bar was observed for the Discover SP-D 35 (CEM Corp.). The described uncontrolled losses of acid can impede the accuracy of the process when volatile metals (e.g. Hg) [67, 68] or metal chlorides (with As, Sb, Sn, etc.) are present [49, 69–72]. Therefore, the MARS 6 turned out to be better suitable for accurate metal analysis in plastic (Table 3) than the Discover SP-D 35 (both CEM Corp.). This conclusion is clearly reflected by the comparison of |*zeta*| scores between both microwave systems (Table 3).

Except for Cr in BAM-H010, all |*zeta*| scores (Eq 1) for the optimized method were well below 2 indicating satisfactory performance (Table 3). With the exception of Cr in BAM-H010 (0.30% ± 0.02%), the recovery rates for the optimized protocol using the MARS 6 fell within a range from 95.9% ± 2.7% (Pb in Lead in Plastic–QC) to 112% ± 7% (*U*; *k* = 2 (*n* = 6)) (Hg in NMIJ CRM 8133-a) (Table 3, S5–S11 Tables in S1 File). The influence of temperature and acid mixtures on the recovery is expanded on in greater detail in Section 3.2 using the example of BAM-H010 (Fig 3).

Reaching a temperature threshold of 230˚C was necessary for the complete dissolution of at least five of the six polymeric matrices. With regard to the PE- (ERM®-E680m and ERM®-EC681m) and PP-based CRMs (NMIJ CRM 8133-a and Lead in Plastic—QC), MWAD at *T* = 210˚C already led to complete dissolution of the materials and acceptable recoveries between 85% and 115%. Complete dissolution of NMIJ CRM 8123-a (PVC) was achieved at a temperature of 230˚C.

Nevertheless, BAM-H010 (ABS) could not be fully digested by means of the MARS 6 microwave system (section 3.2). The occurring yellowish precipitate was identified as 4-nitrobenzoic acid by [1]H NMR spectroscopy (S1 Fig in S1 File). This confirms the result of single crystal x-ray analysis obtained in another study (dealing with a precipitate resulting from incomplete digestion of BAM-H010) [49]. The results stress the demand for methods capable of also

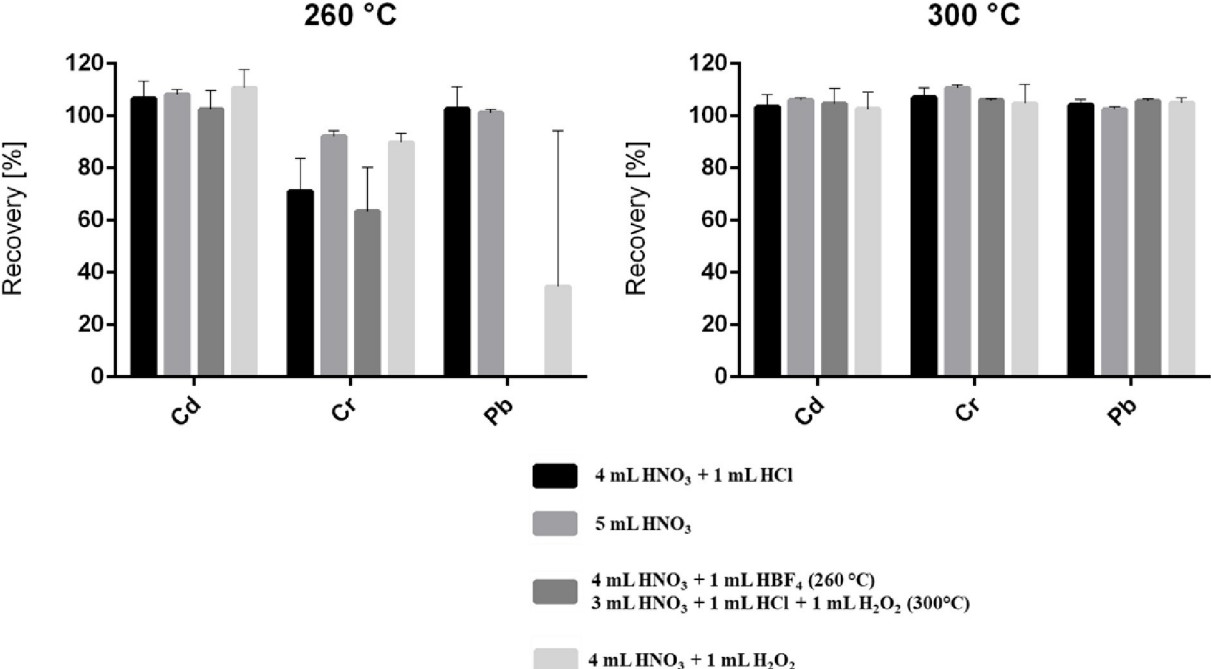

**Fig 3. Recoveries for the digestion of BAM-H010 using the turboWAVE system at 260˚C and 300˚C in conjunction with different reagent mixtures (*U*; *k* = 2 (*n* = 3)).**

mineralizing MP particles consisting of obstinate thermosettings in order to accurately quantify all relevant contained metals. In contrast to thermoplastics (meltable) and elastomers (viscoelasticity), thermosettings show a considerable resistance to thermal and chemical degradation due to the high degree of cross-linking between the polymer chains.

## 3.2 Method optimization using the turboWAVE microwave

Due to the occurring yellowish precipitate (ABS, BAM-H010), additional method optimization was conducted by means of testing a further microwave system. The turboWAVE (MLS GmbH) enables higher digestion temperatures and pressures up to 300˚C and 200 bar, respectively. In order to optimize the recoveries for BAM-H010, especially with regard to Cr (0.30% ± 0.08% ($U$; $k = 2$ ($n = 6$)), the digestion temperature was elevated to (1) 260˚C and (2) 300˚C. The ABS matrix was completely dissolved at 260˚C. Thus, the experiments demonstrated that raising the temperature from 230˚C (section 3.1) to 260˚C—whereby 4 mL $HNO_3$ and 1 mL HCl were used in both cases—improves the recovery for Cr from 0.30% ± 0.08% to 71% ± 25% ($U$; $k = 2$ ($n_1 = 6$; $n_2 = 3$)). The exclusive use of $HNO_3$ even led to a recovery of 93% ± 4% ($U$; $k = 2$ ($n = 3$)) at 260˚C. Recoveries for Cd and Pb were between 100% and 110%. However, none of the three tested reagent mixtures (4 mL $HNO_3$ + 1 mL HCl / 1 mL $HBF_4$ / 1 mL $H_2O_2$) yielded recoveries between 90% and 110% for all three certified elements (Fig 3). Based on the first results, $HBF_4$ was excluded from the subsequent digestions.

When the digestion temperature was raised to 300˚C, all recoveries fell within an range between 102.4% ± 1.8% and 110.5% ± 2.4% ($U$; $k = 2$ ($n = 3$)) regardless of the used reagent mixture. When looking at the four different reagent combinations ($HNO_3$, HCl, and $H_2O_2$), the application of 4 mL $HNO_3$ and 1 mL HCl led to the best results—taking into account both recovery and precision (|$zeta$| = 0.03–0.29). Even though only half of the recommended minimum sample size of the CRM (100 mg instead of 200 mg) was used [73], it was possible to match the certified values with high precision (Fig 3). The difficulties in recovering Cr from BAM-H010 (compared the other plastic CRMs) can be explained by the nature of the metal species present in the material ($Cr_2O_3$). Lethimaki and Väisänen (2017) could also only recover a very small percentage of Cr from BAM-H010 (2.9%) in their study [49]. $Cr_2O_3$ and also $SnO_2$, for instance, are considered virtually not accessible by normal acid digestion methods [74]. Thus, in the replacement of ERM®-EC-680/681k by ERM®-EC-680/681m, $Cr_2O_3$ and $SnO_2$ were substituted by $CaCrO_4$ and $SnS_2$. However, this study has shown that the application of the turboWAVE at a temperature of 300˚C poses an accurate way to dissolve even the most obstinate metal oxides (Fig 3).

## 3.3 Quantification of non-certified elements

Mass fractions for those eight metals, which are certified for the ERM-branded CRMs, were determined for all other CRMs, for which only one to four metals thereof are certified.

Hereby, high mass fractions of Zn were measured in both NMIJ CRMs (Table 3). Likely due to impurities, mass fraction between 0.043 mg kg$^{-1}$ ± 0.027 mg kg$^{-1}$ and 22 mg kg$^{-1}$ ± 7 mg kg$^{-1}$ ($U$; $k = 2$ ($n = 6$)) of Cd, Cr ($w$ = 22 mg kg$^{-1}$ ± 7 mg kg$^{-1}$), Sb, Sn and Zn were detected in the PP matrix of "Lead in Plastic–QC" (trade name of the CRM) (Table 3).

Out of the 48 additionally studied elements (S2 Table in S1 File), the ones, which could be measured (1) at a defined level of relative uncertainty ($U_{rel}$) < 20% and (2) with an acceptable recovery of the quality control standard between 90% and 110%, were selected (Table 4). Mass fractions ranged from 0.016 mg kg$^{-1}$ ± 0.003 mg kg$^{-1}$ (Y in NMIJ CRM 8123-a) to 7.4 g kg$^{-1}$ ± 1.0 g kg$^{-1}$ (Ca in Lead in Plastic–QC) ($U$; $k = 2$ ($n = 6$)).

**Table 4. Mass fractions of selected non-certified elements in the certified reference materials ($U_{rel} <$ 20%; Recovery (QC-Standard) = 90% - 110%).**

| Metal | Mass fraction ($w$ [mg kg$^{-1}$] ($U$; $k$ = 2 ($n$ = 6))) | | | | | | LOQ [mg kg$^{-1}$] |
|---|---|---|---|---|---|---|---|
| | ERM®-E680m | ERM®-E681m | NMIJ CRM 8123-a | NMIJ CRM 8133-a | BAM-H010 | Lead in Plastic—QC | |
| **Al** | 66 ± 8 | 70 ± 6 | < LOQ | 95 ± 10 | 17.6 ± 2.0 | 1000 ± 200 | 5 |
| **Ba** | 3.0 ± 0.4 | 20.8 ± 2.6 | $U_{rel} >$ 20% | $U_{rel} >$ 20% | $U_{rel} >$ 20% | $U_{rel} >$ 20% | 0.04 |
| **Bi** | < LOQ | < LOQ | $U_{rel} >$ 20% | $U_{rel} >$ 20% | 0.200 ± 0.020 | $U_{rel} >$ 20% | $\geq$ 0 |
| **Ca** | $U_{rel} >$ 20% | $U_{rel} >$ 20% | $U_{rel} >$ 20% | < LOQ | 90 ± 8 | 7400 ± 1000 | 4 |
| **Co** | $U_{rel} >$ 20% | 0.55 ± 0.10 | < LOQ | 1.24 ± 0.16 | < LOQ | $U_{rel} >$ 20% | 0.19 |
| **Cu** | 15.9 ± 1.7 | 115 ± 11 | < LOQ | < LOQ | < LOQ | $U_{rel} >$ 20% | 0.14 |
| **Ga** | 0.087 ± 0.019 | 0.59 ± 0.09 | < LOQ | < LOQ | < LOQ | < LOQ | 0.023 |
| **Ge** | < LOQ | < LOQ | < LOQ | < LOQ | < LOQ | 7.5 ± 1.6 | 0.19 |
| **K** | < LOQ | < LOQ | < LOQ | < LOQ | 123 ± 10 | $U_{rel} >$ 20% | 13 |
| **Mg** | $U_{rel} >$ 20% | $U_{rel} >$ 20% | $U_{rel} >$ 20% | 18.2 ± 1.9 | 167 ± 16 | 133 ± 21 | 1.5 |
| **Mo** | < LOQ | $U_{rel} >$ 20% | $U_{rel} >$ 20% | $U_{rel} >$ 20% | $U_{rel} >$ 20% | 0.18 ± 0.04 | 0.005 |
| **Ni** | < LOQ | < LOQ | < LOQ | < LOQ | < LOQ | 1.55 ± 0.23 | 0.08 |
| **Sr** | $U_{rel} >$ 20% | $U_{rel} >$ 20% | $U_{rel} >$ 20% | < LOQ | 0.059 ± 0.013 | 162 ± 24 | 0.05 |
| **Ti** | 3.2 ± 0.5 | 3.2 ± 0.4 | < LOQ | 2.59 ± 0.24 | < LOQ | $U_{rel} >$ 20% | 0.20 |
| **Tl** | < LOQ | $U_{rel} >$ 20% | < LOQ | < LOQ | 0.088 ± 0.010 | 0.030 ± 0.006 | $\geq$ 0 |
| **V** | < LOQ | < LOQ | < LOQ | < LOQ | 0.220 ± 0.028 | 2.1 ± 0.4 | 0.03 |
| **Y** | $U_{rel} >$ 20% | $U_{rel} >$ 20% | 0.016 ± 0.003 | $U_{rel} >$ 20% | < LOQ | $U_{rel} >$ 20% | 0.0007 |

ERM®-EC680m (low level) and ERM®-EC681m (high level) are both based on the same pure LDPE [74, 75]. Therefore, it is likely that Al and Ti (also found in BAM-H010), which mass fractions do not differ significantly between these two CRMs, were contained in the polymer matrix. LDPE is in contrast to HDPE not synthesized by means of a *Ziegler-Natta* catalyst [76, 77]. Therefore, $TiO_2$, used as white opacifier [78] or $TiO_2$ nanoparticles as reinforcing components [79, 80] can be suspected as Ti source. Concerning Al, it has been shown that $Al_2O_3$ is used as a functional filler to enhance LDPE's dielectric resistance [81, 82].

Accordingly, high mass fractions of Al were identified in all CRMs (up to 0.1% *w/w*) except for NMIJ CRM 8123-a. Cu was most probably introduced to the ERM®-EC CRMs via the Pigments 7 and 36, which were employed to also certify Cl and Br mass fractions [74, 75]. The other elements (Ba, Co and Ga) were presumably introduced to the ERM-branded CRMs as impurities in the pigments (oxides and sulfides) used to dope the LDPE with the desired metal mass fractions for certification.

BAM-H010 (ABS) (0.009% *w/w*) and especially Lead in Plastic—QC (PP) ($>$ 0.7% *w/w*) contained higher mass fractions of Ca, which is widely used in the form of $CaCO_3$ as mineral filler in plastic industry [83, 84]. The toxic element Tl was homogeneously contained in two of the CRMs, but at low mass fractions (0.030 mg kg$^{-1}$ ± 0.006 mg kg$^{-1}$–0.088 mg kg$^{-1}$ ± 0.010 mg kg$^{-1}$). While for NMIJ 8123-a (PVC) only Y met the homogeneity criterion ($U_{rel} <$ 20%), 9 further metals were quantifiable in Lead in Plastic—QC. This includes the toxic metal Ni [85], for instance, which is listed in the Water Framework Directive (2000/60/EC) as a priority substance, at a mass fraction of 1.55 mg kg$^{-1}$ ± 0.23 mg kg$^{-1}$ ($U$; $k$ = 2 ($n$ = 6)). Providing information values for metals, for which currently no plastic CRM exists, is helpful for method validation in future studies investigating potential release, sorption and transfer of metal contaminants in aquatic systems mediated by MP particles. For instance, Al, Ni, Co, Ti, Mo and Sr, which could be quantified in the selected CRMs, have already been studied with respect their sorption to MP (Table 1).

## 3.4 Uncertainty evaluation

Combined uncertainties ($u_c$) were calculated for the determination of all certified metals in the representative CRMs ERM®-EC680m and BAM-H010 using the optimized digestion protocol (section 3.1). Resulting relative expanded uncertainties ($U_{rel} = u_{c,rel} \times 2$) ranged from 5% (Cd) to 8% (As) for ERM®-EC680m and from 2.3% (Cd) to 2.7% (Pb) for BAM-H010 (S12 and S13 Tables in S1 File) (Cr excluded for MARS 6 due to neglectable recovery). Fig 4 shows the relative contribution of the type A errors (result of own measurements/observations) stemming from measurement replicates of the instrument and digest replicates as well as of type B errors (result of external sources) such as the certified errors of the scale and the vessels used for digest dilution to the combined uncertainty. Based on the calculation of uncertainties, the identification of the significant sources of uncertainty in the measurement procedure is possible. Furthermore, it shows which parts of the procedure should be handled with care. Improving these parts of the procedure will significantly reduce the overall uncertainty.

Since the main contribution to the combined uncertainty ($> 99\%$) was assigned to the standard deviation of the sextet digest replicates (defined as repeatability conditions) and the standard deviation of the instrument replicates (measurement precision) (Fig 4), the expanded uncertainties ($U; k = 2$) for the non-certified metals were given as double combined standard deviations (Eq 2, Table 4).

$$U(k=2) = 2 \times \sqrt{SD^2_{digest\ replicates} + SD^2_{intrument\ replicates}} \tag{2}$$

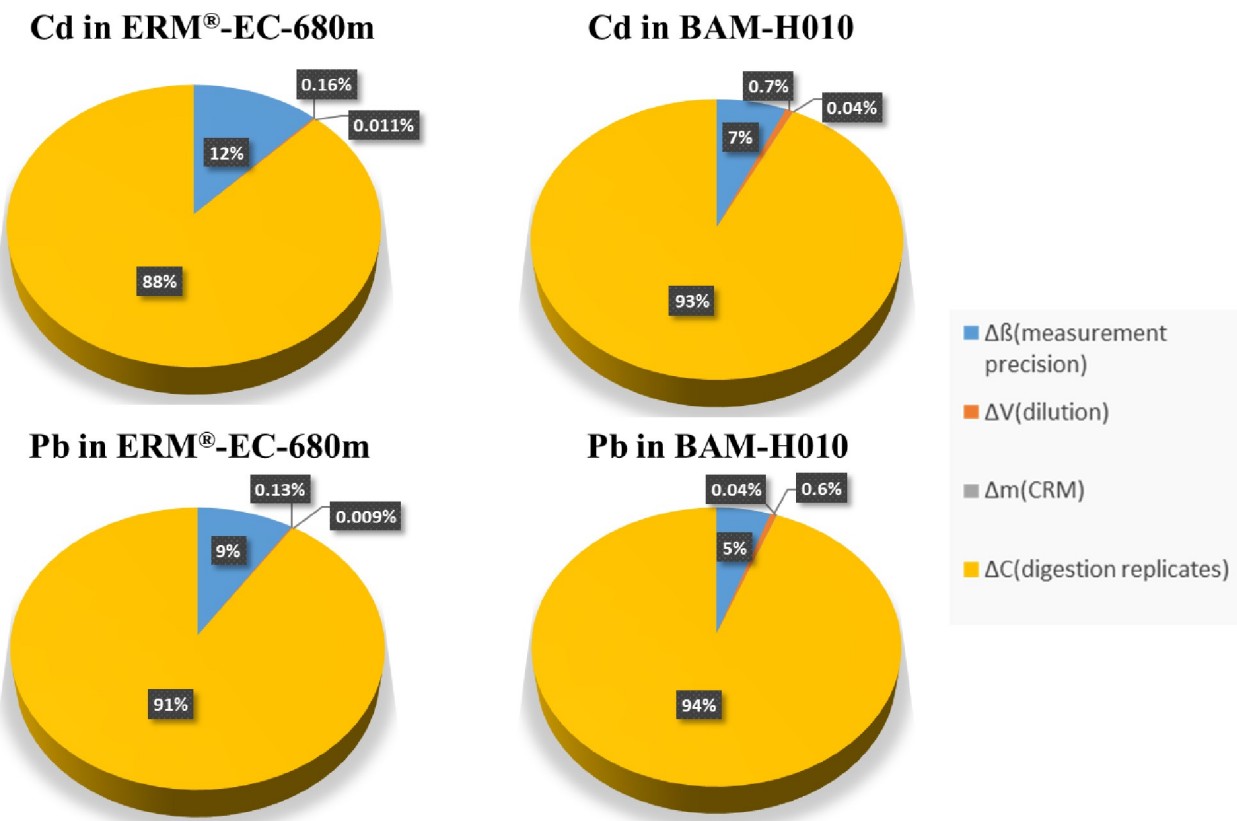

**Fig 4. Relative contribution of the errors of the different input parameters to the overall uncertainty: *Δβ*: Standard deviation of ICP-MS/MS measurement replicates; *ΔV*: Error of the dilution of the digests; *Δm*: Error of the scale; *ΔC*: Standard deviation of the concentrations measured in the digest replicates.**

Eq 2: Calculation of the expanded uncertainty.

However, the reported uncertainties do not take into account reproducibility conditions, e.g. different principle of measurement, measuring instrument or location (GUM (JCGM 100:2008)) [39]. Interlaboratory tests organized and evaluated by the Swiss Federal Laboratories for Materials Science and Technology (2004) have already given valuable insights in the reproducibility limits ($2.8 \times (SD_{Repeatability} + SD_{Interlaboratory})^{0.5}$) of heavy metal determinations in different polymer matrices (12% - 65% for Pb, Cr and Cd in PU and PVC) [86].

## 4. Summary and outlook

The present study pursued three main goals:

i.  Provision of MWAD protocols suitable for trace metal analysis on/in MP particles of different plastic types (Table 5)

ii.  Provision of data on non-certified present metals in the selected CRMs

iii.  Experimental demonstration of the complexity of metal analysis in the material class of synthetic polymers as a highly heterogeneous pool of matrices

A first optimized protocol yielded good recoveries from 95.9% ± 2.7% to 112% ± 7% ($U$; $k = 2$ ($n = 6$)) for As, Cd, Cr, Hg, Sn, Sb and Zn for six different plastic CRMs (PE, PP, PVC, ABS) ($T = 230°C$). The low recovery for Cr in ABS (BAM-H010) (0.30% ± 0.05%) constituted the only outlier. Further method optimization by means of a more powerful microwave system ($T = 300°C$) led to considerable improvements of recovery and precision for the challenging ABS CRM (103% ± 10% - 107 ± 8% ($U$; $k = 2$ ($n = 3$)). ABS is of high relevance as it is a major constituent of electrotechnical waste which can contain high amounts of heavy metals [55]. Therefore, for studies focusing on electrotechnical waste and other very obstinate plastic matrices, we recommend the use of the 2$^{nd}$ proposed protocol (Table 5).

Our findings experimentally underpin the complexity of metal analysis in different polymeric matrices, which has also to be recognized by all scientists analyzing the interactions between metals and MP particles. Therefore it is mandatory to stick to validated protocols applied by other scientific fields, such as materials science and environmental analytical chemistry. Even if weak leaching protocols to extract adsorbed metals are applied (Table 1), validation on the basis of comparison to a complete digestion is indispensable to evaluate the influence of the bulk metal mass fractions.

Table 5. Recommendation of two MWAD protocols for metal analysis in (particulate) plastic depending on the research question and the available microwave system.

| No. | (Particulate) plastic of interest | Chemical and heat resistance | Solubility of metal species | Temperature [° C] (microwave systems) | Acid mixture ($V$ [mL]) | $m$ (plastic) [mg] | CRM (polymer type) | Ramp time [min] | Hold time [min] |
|---|---|---|---|---|---|---|---|---|---|
| 1 | Most common synthetic polymer materials | Low—normal | Normal | 230 (most commercial MW systems) | HNO$_3$ (4), HCl (1) | 60–100 | ERM-EC680m (PE) and NMIJ 8133-a (PP) | 20 | 60 |
| 2 | Specialty/ high performance polymers | High | Very low | 300 (only special MW systems) | HNO$_3$ (4), HCl (1) | 200 | BAM-H010 (ABS) | 20 | 30 |

This knowledge should be taken into account for future analysis of the interactions between metal and particulate plastic contaminants in the aquatic environment. Our main conclusion is that only the application of validated analytical procedures (based on matrix-matched CRMs) generates comparable and significant data on the role of MP as a vector for metals.

For future studies investigating the interactions (sorption and desorption processes and the inherent metal content) between the most common environmental particulate plastic types and metal contaminants, we recommend the use of the 1st proposed MWAD protocol (Table 5) for total acid digestion.

## Supporting information

**S1 File.**
(DOCX)

## Acknowledgments

We especially thank Prof. Dr. Frank Sönnichsen (University of Kiel) for providing the NMR spectrometer.

## Author Contributions

**Conceptualization:** Lars Hildebrandt, Marcus von der Au, Tristan Zimmermann, Daniel Pröfrock.

**Data curation:** Lars Hildebrandt.

**Formal analysis:** Lars Hildebrandt.

**Investigation:** Lars Hildebrandt, Marcus von der Au, Jannis Ludwig.

**Methodology:** Lars Hildebrandt, Tristan Zimmermann, Anna Reese, Daniel Pröfrock.

**Resources:** Daniel Pröfrock.

**Software:** Lars Hildebrandt.

**Supervision:** Daniel Pröfrock.

**Validation:** Lars Hildebrandt.

**Visualization:** Lars Hildebrandt.

**Writing – original draft:** Lars Hildebrandt, Marcus von der Au.

**Writing – review & editing:** Lars Hildebrandt, Marcus von der Au, Tristan Zimmermann, Anna Reese, Daniel Pröfrock.

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
