## [Decision Letter · Decision Letter 0]

22 May 2020

PONE-D-20-09074

Microwave-assisted acid digestion protocol for accurate trace metal analysis in different types of microplastic

PLOS ONE

Dear Mr. Hildebrandt,

Thank you for submitting your manuscript to PLOS ONE. After careful consideration, we feel that it has merit but does not fully meet PLOS ONE’s publication criteria as it currently stands. Therefore, we invite you to submit a revised version of the manuscript that addresses the points raised during the review process.

three independent referees commented on your submission. All of them find the submission of scientific value, and made concrete and detailed suggestions for improving the manuscript.

I am looking forward to receive your revised version.

All the very best greetings,

Martin Koller

We would appreciate receiving your revised manuscript by Jul 06 2020 11:59PM. To enhance the reproducibility of your results, we recommend that if applicable you deposit your laboratory protocols in protocols.io, where a protocol can be assigned its own identifier (DOI) such that it can be cited independently in the future. For instructions see: http://journals.plos.org/plosone/s/submission-guidelines#loc-laboratory-protocols

We look forward to receiving your revised manuscript.

Kind regards,

Martin Koller, Ph.D.

Academic Editor

PLOS ONE

Journal Requirements:

Reviewers' comments:

Reviewer's Responses to Questions

**Comments to the Author**

1. Is the manuscript technically sound, and do the data support the conclusions?

Reviewer #1: Yes

Reviewer #2: Yes

Reviewer #3: Yes

2. Has the statistical analysis been performed appropriately and rigorously? 

Reviewer #1: Yes

Reviewer #2: Yes

Reviewer #3: Yes

3. Have the authors made all data underlying the findings in their manuscript fully available?

Reviewer #1: Yes

Reviewer #2: Yes

Reviewer #3: Yes

4. Is the manuscript presented in an intelligible fashion and written in standard English?

Reviewer #1: Yes

Reviewer #2: Yes

Reviewer #3: Yes

5. Review Comments to the Author

Reviewer #1: The article present a research work on the development of a standard protocol for the analysis of metals which are sorbed at microplastic particles. The general aim of this work, the establishment of standardised procedures and the use of reference materials is highly welcome and deserves high attention, especially in the area of microplastic research.

However in my eyes at several point the real impact of such a work will not really reach the audience. Therefor the authors must make a greater effort to communicate the metrological-analytical value of such work to a community that often does not take such aspects into account when answering their own questions.

I will this comment at specific point

L1: The title does not really meet the aspect of the work, because its just summarise the technical part, which was done. Please think about a title more related to the outcome of the work, like: “Provision of a harmonized/standard/metrologically traceable protocol for the analysis of sorbed metal ions on microplastic particles”

L60/61: The sentence “Two aspects are important for the interactions between metals and MP” is misleading, because the two described aspects are not “imported”, they are the aspects in the debate. Better would be something like “Two scenarios are currently being considered in this context”.

L 79-86: The use of CRM must be highlighted stronger. Please make clearer, that the existing studies gave very different results and no harmonised procedures were used. Make clear, that for meaningful assessment of the data harmonised protocols are needed, including reference materials or inter-laboratory comparison test. Make clear, what an uncertainty in measurements means. For somebody from metrology this sounds simple, but the MP community often do not know those aspects. At this point of the paper the authors “have to pick up the customers of their work”.

L101-103: Please give also a reason for using those “exemplary chosen materials” with relation to microplastic findings.

L105-110: This explanation is right but must not explained here, please delete it, this information is not relevant for the article. ABS is used here as a representative of styrene copolymers.

L111-114: The sentence does not hit the core of the work “Therefore, the aim of this study is to contribute to a better understanding of the interactions between MP and metal contaminants by providing a thoroughly validated polymer-specific MWAD protocol for metal analysis in MP.”

The core is “Therefore, the aim of this study is to provide a thoroughly validated polymer-specific MWAD protocol for metal analysis in MP for a better understanding of the interactions between MP and metal contaminants”.

Figure 1: Please add also polymer specific parameters in the figure, like glass transition temperature, or crystallisation content which are of the most relevant parameters for sorption studies (See A. Müller, R. Becker, U. Dorgerloh, F.-G. Simon, U. Braun, Environmental Pollution 2018, 240, 639-646.). Even when this is not relevant for the present work, it would be helpful to add is aspect in the discussion.

Figure 2: The meaning of the figure does not reach me. What is the need for this kind of figure in this work?

L 243-244: The sentence “The results stress the demand for methods capable of also mineralizing MP particles consisting of obstinate thermosettings in order to accurately quantify all relevant contained metals.” address new aspect, which could confuse the reader. Please made the sentence more general or explain, what is the difference in chemical structure (and therefore mineralisation process) of thermosets, thermoplastics and elastomers.

Figure 4: The meaning of the figure does not reach me, what is delta beta, delta V, delta M and delta C? The figure seems established for metrology, but not for the audience of MP community. Please explain the parameters and the achieved information of them more detailed.

L 359/360 “Demonstration and discussion of the complexity of metal analysis in the material class of synthetic polymers as a highly heterogeneous pool of matrices”. A discussion about synthetic polymers as a highly heterogeneous pool of matrices was not really done, because the polymer chemistry and properties were not discussed. It was just a demonstration.

L 374-376: “Finally, only by a combination of bulk methods and specific surface-analysis techniques such as LA-ICP-MS or μ-XRF, the question “How much of the associated metal is bound to the surface and how much is contained in the polymer matrix of MP particles?” can be tackled. Answering this questions enables assessment of the availability of associated metal contaminants for organisms ingesting MP.”

The article presence a harmonised protocol for metal analysis. The article does not address analysis of different sorption processes, like adsorption or absorption. Actually, regarding the size of some of the metals ions and the existing free volume in a polymer an absorption is less likely. For a meaningful comment on this, please implements also aspects of permeation, diffusion coefficient, solubility etc. Otherwise, a question is raised here which has already been completely misunderstood in the MP community and does not draw on existing polymer knowledge enough. The results of using bulk or surface sensitive methods are negligible if polymers above or below the Tg are studied, various crystallization grades are not considered or the chemical structure of the polymer…

Therefore, please also change the abstract: The high-surface-to-volume-ratio must not be the relevant parameter for sorption process!

Finally comment

Could you please give a “recommendation” for the user of the protocol, which auf the materials are most promising to use them as a reference materials for own investigations?

Reviewer #2: The manuscript is an important contribution to improving analytics with respect to microplastics. Which is appreciated, because quite some work done in microplastics suffers from poor analytical methods …. It is easy to read and well written. I hence only have some minor comments which will, I hope, increase its readability further.

Abstract: “… using six different certified reference materials in the MP size range, i.e. ERM®-EC680m, ERM®-EC681m, NMIJ CRM 8123-a, NMIJ CRM 8133-a, BAM-H010 and Lead in Plastic - QC.” Very few readers will understand these ‘code names’ of the reference materials and their listing will hence tend to discourage a non-nerd within the field from reading the article. It is fully sufficient for an abstract to write e.g. “… using six different certified reference materials in the MP size range consisting of polyethylene, polypropylene, acrylonitrile butadiene styrene and polyvinyl chloride”. The same change goes for the rest of the abstract, avoid these ‘code names. (also note that the plastics are not trade names and should not capitalized)

Line 85-86: “calculation of uncertainty budgets according to the “Guide to the expression of uncertainty in measurement” (GUM (JCGM 100:2008)).” Please give a proper reference to this guideline. The text in the brackets is not a proper reference. This phrase also pops up later in the text. Also here it needs a reference.

Line 118-119: “All experiments based on the MARS 6 and the Discover SP-D 35 (CEM Corp., Kamp Lintfort, Germany) microwaves …”. What are MARS 6 microwaves? What is Discover SP-D 35 microwaves? Are these some special sort of microwave radiation? (I know they are digestion systems, but a reader cannot see that from the text). Something is wrong in this sentence, a reader not familiar with these machines? / techniques? will not understand this sentence.

Line 130: what is meant by “(Lead in plastic - QC)”? This phrase is used later in the text also, but I have not been able to figure out what is meant hereby.

Various places in the manuscript: Note that the word “lead” (or other metals or substances) should not be capitalized hence lead, not Lead.

Various places in the manuscript: Note that the word “plastic” should not be capitalized hence plastic, not Plastic.

Reviewer #3: The manuscript submitted by Hildebrandt and co-workers describes the development of a microwave-heated acid digestion protocol for the trace metal analysis in microplastics. In order to gain a better understanding of metal contaminants in microplastics and their accurate analysis, six different certified reference materials (comprising the most important polymeric matrices such as PE, PP and PVC) were analyzed for their metal content by ICP-MS/MS. In addition, non-certified elements in the CRMs were quantified. In times where the discussion about body uptake of microplastics and thus the within contained (toxic) substances is heavily ongoing, this study puts an additional focus to this field.

The manuscript is well written and constructed, and analyses (as far as I can rate) well performed. However, what I miss, since the authors claim a thorough microwave-heated acid digestion protocol, is exactly this protocol. In their Experimental they report different temperatures and times that have been used for the MW protocol using 2 different MW instruments (MARS 6 and Discover SP-D 35). Also different digestion chemicals are listed. In the Results and Discussion section, only data for the MARS 6 MW is discussed, whereas additional data is provided in Table 3 for both instruments. But the authors do not explain why different acid mixtures have been used, and if the temperatures and times for both instruments were the same. Also the method of temperature measurement is not given (internal or external). I would like to know why 2 different instruments with different digesting mixtures have been used, what are the limitations of those instruments with respect to the turboWAVE and what would be the best digestion protocol for the CRMs. Can a general protocol be applied, or does the protocol need to be adapted depending on the CRM and type of MW instrument, respectively? A bit more detailled information on such a protocol would be desirable, also with respect to reproducibility.

Minor comments:

1) Which cleaning procedure was applied for the MW vials?

2) page 8, line 169: specify TFM

3) page 8, line 170: I would suggest a rewording of "CRMs were submerged with ..." to "The respective amount of ... was added to the MW vessels containg the CRMs"

4) page 11, line 240: it should be 4-nitrobenzoic acid

5) page 15, line 276: ...be explained by the presence of the metal species... I would rather say: ...be explained by the nature of the metal species...

6) page 19, line 327: I would recommend an ascending labelling of the SI Tables according to the text. Tables A5+A6 appear last in the text but not in the SI

7) page 19, Figure 4: make the pie chart clearer for Pb in BAM-H010, one does not clearly see to which legend the 0.04% and 0.6% belong

8) Figure A1 in the SI: I do not see the acidic proton at 13.30 ppm in the 1H NMR spectrum!!! I also don´t think that according only to that NMR, the structure of 4-nitrobenzoic acid can be confirmed.

6. PLOS authors have the option to publish the peer review history of their article (what does this mean?). If published, this will include your full peer review and any attached files.

Reviewer #1: No

Reviewer #2: Yes: Jes Vollertsen

Reviewer #3: No

---

## [Author Response · Author response to Decision Letter 0]

18 Jun 2020

Reviewer #1: 

Comment/Question: The article present a research work on the development of a standard protocol for the analysis of metals which are sorbed at microplastic particles. The general aim of this work, the establishment of standardised procedures and the use of reference materials is highly welcome and deserves high attention, especially in the area of microplastic research.

However in my eyes at several point the real impact of such a work will not really reach the audience. Therefor the authors must make a greater effort to communicate the metrological-analytical value of such work to a community that often does not take such aspects into account when answering their own questions.

Answer: Thank you very much for the valuable and helpful input! We tried to address all of it to transport main messages in a better way (the use of CRMs by the MP community is necessary to improve the data, metrology concepts established in analytical chemistry have to be taken into account by the MP community). The modifications and additions are visible in the track changes version of the revised manuscript.

Comment/Question: I will this comment at specific point

L1: The title does not really meet the aspect of the work, because its just summarise the technical part, which was done. Please think about a title more related to the outcome of the work, like: “Provision of a harmonized/standard/metrologically traceable protocol for the analysis of sorbed metal ions on microplastic particles”

Answer: L1-4: “A metrologically traceable protocol for the quantification of trace metals in different types of microplastic” (we do not want to mention sorption/sorbed… in the title, since the protocol can be applied (also by other fields). It is suitable whenever someone wants to analyze the metal content of plastic (materials science, recycling sector etc.)

Comment/Question: L60/61: The sentence “Two aspects are important for the interactions between metals and MP” is misleading, because the two described aspects are not “imported”, they are the aspects in the debate. Better would be something like “Two scenarios are currently being considered in this context”.

Answer: L68 - L 69: The mentioned section has been changed to “Two scenarios are currently being considered in this context”.

Comment/Question: L 79-86: The use of CRM must be highlighted stronger. Please make clearer, that the existing studies gave very different results and no harmonised procedures were used. Make clear, that for meaningful assessment of the data harmonised protocols are needed, including reference materials or inter-laboratory comparison test. Make clear, what an uncertainty in measurements means. For somebody from metrology this sounds simple, but the MP community often do not know those aspects. At this point of the paper the authors “have to pick up the customers of their work”.

Answer: L 91 – 111: The following section has been modified: “Application of such non- or poorly (according to international metrology standards) validated procedures leads to generation of inaccurate, non-traceable and incomparable data. Therefore, in analytical chemistry, using matrix-matched CRMs is indispensable for the generation of comparable and metrologically traceable data as well as the calculation of uncertainty budgets according to the “Guide to the expression of uncertainty in measurement” (GUM (JCGM 100:2008)) [39]. The formal definition of “uncertainty of measurement” would be: “parameter, associated with the result of a measurement, that characterizes the dispersion of the values that could reasonably be attributed to the measurand” [39] (measurand in this context may be replaced with concentration for most areas of chemical analysis).

Expanded uncertainties take into account all major potential error contributions (e.g. measurement precision, reproducibility, inhomogeneity of the sample, blank contribution) (Figure 2) and a coverage factor (in the case of assumed normal distribution using ± two combined uncertainties refers to a 95.4% confidence interval). Therefore, uncertainties will not only give a measure of the quality of a result enabling the user to assess the reliability of analytical data, they also facilitate identification of the significant sources of uncertainty in a measurement procedure. Only if there is no overlap of the referring confidence intervals of two means, effects are significant based on a predefined significance level (α). For meaningful assessment of the data on the interactions between metals and MP but also for data on the general abundance of MP particles and fibers [40], thorough method validation and harmonized protocols are needed, including reference materials, inter-laboratory comparison tests and sound applications of existing metrological-analytical concepts.”

Comment/Question: L101-103: Please give also a reason for using those “exemplary chosen materials” with relation to microplastic findings.

Answer: The following section has been modified L135 – 136: … and a high share of the MP particles typically detected in aquatic environments [51-53]

Comment/Question: L105-110: This explanation is right but must not explained here, please delete it, this information is not relevant for the article. ABS is used here as a representative of styrene copolymers.

Answer: The following section has been modified L 137 - 138 “…acrylonitrile butadiene styrene (ABS)…as a representative of styrene copolymers…”; L140 - 146: deleted

Comment/Question: L111-114: The sentence does not hit the core of the work “Therefore, the aim of this study is to contribute to a better understanding of the interactions between MP and metal contaminants by providing a thoroughly validated polymer-specific MWAD protocol for metal analysis in MP.”

The core is “Therefore, the aim of this study is to provide a thoroughly validated polymer-specific MWAD protocol for metal analysis in MP for a better understanding of the interactions between MP and metal contaminants”.

Answer: The following section has been modified L146 – 149: “The aim of this study is to provide a thoroughly validated polymer-specific MWAD protocol for metal analysis in MP for a better understanding of the interactions between MP sampled in different environments and metal contaminants.”; the rest was deleted.

Comment/Question: Figure 1: Please add also polymer specific parameters in the figure, like glass transition temperature, or crystallisation content which are of the most relevant parameters for sorption studies (See A. Müller, R. Becker, U. Dorgerloh, F.-G. Simon, U. Braun, Environmental Pollution 2018, 240, 639-646.). Even when this is not relevant for the present work, it would be helpful to add is aspect in the discussion.

Answer: We really appreciate this point. We tried to only discuss parameters that can really have a negative influence on the recovery in “complete acid digestion processes”. In this scenario, only obstinate metal species can pose a problem. However, our paper tackles your important aspect now in the introduction part on “weak acidic extraction/leaching protocols” because for these protocols the strength of sorption is of high importance:

Also see Abstract: L 51 – 52: “Addressing specific analysis tools for different sorption scenarios and processes as well as the underlying kinetics was beyond this study’s scope.”

In addition the following section has been modified L 113 – 122: “Additionally, the degree of desorption (achieved by leaching) can vary between different polymer types (depending on the chemical structure of the polymeric chain). Maybe even more importantly, a meaningful assessment of the sorption and desorption behavior cannot be conducted without considering a variety of physical parameters, e.g. permeability, diffusion coefficients, solubility and polarity [41]. Müller et al. (2018) have demonstrated that sorption (and herewith also desorption) of chemicals to MP is highly influenced by polymer-specific parameters such as glass transition temperature and crystallization content [42]. 

To overcome resulting selectivity differences, it is advisable to put future studies focusing on the role of MP as a vector for metal contaminants either on the basis of a complete microwave-assisted acid digestion (MWAD) protocol …”

Comment/Question: Figure 2: The meaning of the figure does not reach me. What is the need for this kind of figure in this work?

Answer: This Figure should explain in a visual way, how combined uncertainties and the resulting expanded uncertainties (as a central aspect of the publication to be discussed) are generated based on the entire analytical process: 

The following section has been modified L260 – L261: “All error contributing to the overall uncertainty must be considered.”

The following section has been modified L 100 – 102: “Expanded uncertainties take into account all major potential error contributions (e.g. measurement precision, reproducibility, inhomogeneity of the sample, blank contribution) (Figure 2) and a coverage factor…”

Comment/Question: L 243-244: The sentence “The results stress the demand for methods capable of also mineralizing MP particles consisting of obstinate thermosettings in order to accurately quantify all relevant contained metals.” address new aspect, which could confuse the reader. Please made the sentence more general or explain, what is the difference in chemical structure (and therefore mineralisation process) of thermosets, thermoplastics and elastomers.

Answer: The following section has been modified L 313 – 317: “The results stress the demand for methods capable of also mineralizing MP particles consisting of obstinate thermosettings in order to accurately quantify all relevant contained metals. In contrast to thermoplastics (meltable) and elastomers (viscoelasticity), thermosettings show a considerable resistance to thermal and chemical degradation due to the high degree of cross-linking between the polymer chains.”

Comment/Question: Figure 4: The meaning of the figure does not reach me, what is delta beta, delta V, delta M and delta C? The figure seems established for metrology, but not for the audience of MP community. Please explain the parameters and the achieved information of them more detailed.

Answer: L 401 – 405: “Figure 4 shows the relative contribution of the type A errors (result of own measurements/observations) stemming from measurement replicates of the instrument and digest replicates as well as of type B errors (result of external sources) such as the certified errors of the scale and the vessels used for digest dilution to the combined uncertainty. Based on the calculation of uncertainties, the identification of the significant sources of uncertainty in the measurement procedure is possible. Furthermore, it shows which parts of the procedure should be handled with care. Improving these parts of the procedure will significantly reduce the overall uncertainty.”

The following section has been modified L 411 – 414: “Figure 4: Relative contribution of the errors of the different input parameters to the overall uncertainty: Δβ: standard deviation of ICP MS/MS measurement replicates; ΔV: error of the dilution of the digests; Δm: error of the scale; ΔC: standard deviation of the concentrations measured in the digest replicates.”

Comment/Question: L 359/360 “Demonstration and discussion of the complexity of metal analysis in the material class of synthetic polymers as a highly heterogeneous pool of matrices”. A discussion about synthetic polymers as a highly heterogeneous pool of matrices was not really done, because the polymer chemistry and properties were not discussed. It was just a demonstration.

Answer: Thank you, you are definitely right! We wanted to keep a clear focus on the methodological difficulties of metal analysis in plastic. We will tackle the sorption and desorption processes/important aspects in a future study specifically dealing with sorption and desorption (based on our digestion protocol) experiments since it is of enormous complexity.

The following section has been modified L 442/443: “III. Experimental demonstration of the complexity of metal analysis in the material class of synthetic polymers as a highly heterogeneous pool of matrices”

Comment/Question: L 374-376: “Finally, only by a combination of bulk methods and specific surface-analysis techniques such as LA-ICP-MS or μ-XRF, the question “How much of the associated metal is bound to the surface and how much is contained in the polymer matrix of MP particles?” can be tackled. Answering this questions enables assessment of the availability of associated metal contaminants for organisms ingesting MP.”

The article presence a harmonised protocol for metal analysis. The article does not address analysis of different sorption processes, like adsorption or absorption. Actually, regarding the size of some of the metals ions and the existing free volume in a polymer an absorption is less likely. For a meaningful comment on this, please implements also aspects of permeation, diffusion coefficient, solubility etc. Otherwise, a question is raised here which has already been completely misunderstood in the MP community and does not draw on existing polymer knowledge enough. The results of using bulk or surface sensitive methods are negligible if polymers above or below the Tg are studied, various crystallization grades are not considered or the chemical structure of the polymer…

Therefore, please also change the abstract: The high-surface-to-volume-ratio must not be the relevant parameter for sorption process!

Answer: Thank you again for this very important hint! Our main message is, that it is mandatory to use polymer CRMs for all future studies investigating the “interactions” between MP and metals in whatever way. You are totally right, that we are not presenting a sorption study here. In order to keep our message maximally clear, we do not want to discuss the relevant parameters for sorption (ad- and absorption) and desorption (in the discussion section). This should be done either in a review or a future paper on sorption and desorption (in which we have gained first experimental insights yet, on which we have to expand on in the future).

The following section has been modified L29: “…high surface to volume ratio…” � deleted

The following section has been modified L 51 – 56: “… Addressing specific analysis tools for different sorption scenarios and processes as well as the underlying kinetics was beyond this study’s scope. However, the future application of the two recommended thoroughly validated total acid digestion protocols as a first step in the direction of harmonization of metal analysis in/on MP will enhance the significance and comparability of the generated data. It will contribute to a better understanding of the role of MP as vector for trace metals in the environment.”

The following section has been modified L 464 – 468: deleted

The following section has been modified: L 113 – 122: “Additionally, the degree of desorption (achieved by leaching) can vary between different polymer types (depending on the chemical structure of the polymeric chain). Maybe even more importantly, a meaningful assessment of the sorption and desorption behavior cannot be conducted without considering a variety of physical parameters, e.g. permeability, diffusion coefficients, solubility and polarity [41]. Müller et al. (2018) have demonstrated that sorption (and herewith also desorption) of chemicals to MP is highly influenced by polymer-specific parameters such as glass transition temperature and crystallization content [42]. To overcome resulting selectivity differences, it is advisable to put future studies focusing on the role of MP as a vector for metal contaminants either on the basis of a complete microwave-assisted acid digestion (MWAD) protocol or…”

Comment/Question: Finally comment

Could you please give a “recommendation” for the user of the protocol, which auf the materials are most promising to use them as a reference materials for own investigations?

Answer: Thank you! This point was addressed by the other Reviewers as well. The readers that want to apply and reproduce the applied protocol should see it more easily. Therefore, a table was added. 

Table 5

The following section has been modified L 450 – 452: “…Therefore, for studies focusing on electrotechnical waste and other very obstinate plastic matrices, we recommend the use of the 2nd proposed protocol (Table 5).”

The following section has been modified L 469 – 472: “For future studies investigating the interactions (sorption and desorption processes and the inherent metal content) between the most common environmental particulate plastic types and metal contaminants, we recommend the use of the 1st proposed MWAD protocol (Table 5) for total acid digestion.”

Reviewer #2: 

Comment/Question: The manuscript is an important contribution to improving analytics with respect to microplastics. Which is appreciated, because quite some work done in microplastics suffers from poor analytical methods …. It is easy to read and well written. I hence only have some minor comments which will, I hope, increase its readability further.

Answer: Thank you very much for the important and constructive input which hopefully helps us to reach a broader audience within the microplastic community! The modifications and additions are visible in the track changes version of the revised manuscript.

Comment/Question: Abstract: “… using six different certified reference materials in the MP size range, i.e. ERM®-EC680m, ERM®-EC681m, NMIJ CRM 8123-a, NMIJ CRM 8133-a, BAM-H010 and Lead in Plastic - QC.” Very few readers will understand these ‘code names’ of the reference materials and their listing will hence tend to discourage a non-nerd within the field from reading the article. It is fully sufficient for an abstract to write e.g. “… using six different certified reference materials in the MP size range consisting of polyethylene, polypropylene, acrylonitrile butadiene styrene and polyvinyl chloride”. The same change goes for the rest of the abstract, avoid these ‘code names. (also note that the plastics are not trade names and should not capitalized)

Answer: The abstract has been adapted. Trade names were replaced by polymer types (L 38 - 39, L 45, L 49)

The following section has been modified L 35 – 37: “…using six different certified reference materials in the microplastic size range consisting of polyethylene, polypropylene, acrylonitrile butadiene styrene and polyvinyl chloride.”

Comment/Question: Line 85-86: “calculation of uncertainty budgets according to the “Guide to the expression of uncertainty in measurement” (GUM (JCGM 100:2008)).” Please give a proper reference to this guideline. The text in the brackets is not a proper reference. This phrase also pops up later in the text. Also here it needs a reference.

Answer: The reference was added: L96, 98, 265, 425 - 426, 611 - 612

Comment/Question: Line 118-119: “All experiments based on the MARS 6 and the Discover SP-D 35 (CEM Corp., Kamp Lintfort, Germany) microwaves …”. What are MARS 6 microwaves? What is Discover SP-D 35 microwaves? Are these some special sort of microwave radiation? (I know they are digestion systems, but a reader cannot see that from the text). Something is wrong in this sentence, a reader not familiar with these machines? / techniques? will not understand this sentence.

Answer: L 159 – 174: “The three microwave systems compared in this study differ in the general construction, but the main practical differences refer to the number of vessels that can be processed at a time, the vessel sizes (section 2.2) and the pressure as well as temperature regulation. Briefly summarized, the MARS 6 and the Discover SP-D 35 (external IR temperature control; pressure vessels) used in this study enable digestion at temperatures up to 230 °C and observed pressures up to 24 - 28 bar, whereas the turboWAVE bears a significantly higher maximum temperature of 300 °C and also a significantly higher maximum pressure of 200 bar. In the tuboWAVE, Temperature and pressure are both regulated and controlled in a single reaction chamber filled with inert gas. In contrast to the MARS 6 and the turboWAVE microwave, that feature simultaneous processing of a batch of digestions (40 and 15 vessels), the Discover SP D 35 (in conjunction with an Explorer autosampler) irradiates the vessels automatically one after another enabling variation of digestion parameters for method development (different conditions for every vessel possible). Please note that this comparison is not meant to be exhaustive, since there are a lot of different vessel types (e.g. for different maximum pressures and temperatures), add-ons features (e.g. for pressure and temperature control) and also other microwave systems available on the market.”

Comment/Question: Line 130: what is meant by “(Lead in plastic - QC)”? This phrase is used later in the text also, but I have not been able to figure out what is meant hereby.

Answer: This is just the trade name, like ERM-EC680m:

https://www.sigmaaldrich.com/catalog/product/sial/sqc1093?lang=de&region=DE

L183 – 184 …and another PP CRM (Lead in plastic – QC (trade name)) from Sigma Aldrich (Wyoming, USA)…

L 362: …’“Lead in Plastic – QC” (trade name of the CRM) (Table 3).’

Comment/Question: Various places in the manuscript: Note that the word “lead” (or other metals or substances) should not be capitalized hence lead, not Lead.

Answer: See previous point, in this case it is part of the trade name.

Comment/Question: Various places in the manuscript: Note that the word “plastic” should not be capitalized hence plastic, not Plastic.

Answer: Plastic is only capitalized when it is part of “Lead in Plastic – QC”

Reviewer #3: 

Comment/Question: The manuscript submitted by Hildebrandt and co-workers describes the development of a microwave-heated acid digestion protocol for the trace metal analysis in microplastics. In order to gain a better understanding of metal contaminants in microplastics and their accurate analysis, six different certified reference materials (comprising the most important polymeric matrices such as PE, PP and PVC) were analyzed for their metal content by ICP-MS/MS. In addition, non-certified elements in the CRMs were quantified. In times where the discussion about body uptake of microplastics and thus the within contained (toxic) substances is heavily ongoing, this study puts an additional focus to this field.

Answer: Thank you very much. We really appreciate the valuable and constructive input and we have tried to address all of it thoroughly! The modifications and additions are visible in the track changes version of the revised manuscript.

Comment/Question: The manuscript is well written and constructed, and analyses (as far as I can rate) well performed. However, what I miss, since the authors claim a thorough microwave-heated acid digestion protocol, is exactly this protocol. In their Experimental they report different temperatures and times that have been used for the MW protocol using 2 different MW instruments (MARS 6 and Discover SP-D 35). Also different digestion chemicals are listed. In the Results and Discussion section, only data for the MARS 6 MW is discussed, whereas additional data is provided in Table 3 for both instruments. But the authors do not explain why different acid mixtures have been used, and if the temperatures and times for both instruments were the same. 

Answer: We added a section on the choice of the preferred acid mixture and an explanation why the MARS 6 is considered more suitable than the Discover SP-D 35 (in addition to the reflection by zeta scores in table 3)

The following section has been modified L 285 – 295: “The use of sulfuric acid (H2SO4) was omitted in order to avoid non-spectral interferences [72] and reduced recoveries for Pb, due to the formation of insoluble sulfates [47]. In General, the addition of H2O2 and HBF4 to the mixture of HNO3 and HCl did not lead to better recoveries (Figure 3) but - in case of H2O2 - an increase of pressure in the microwave vessels and higher losses of acid mixture during the digestion were observed. Moreover, an automatic release of the vessel pressure above approximately 24 bar was observed for the Discover SP-D 35 (CEM Corp.). The described uncontrolled losses of acid can impede the accuracy of the process when volatile metals (e.g. Hg) [73, 74] or metal chlorides (with As, Sb, Sn, etc.) are present [49, 75-78]. Therefore, the MARS 6 turned out to be better suitable for accurate metal analysis in plastic (Table 3) than the Discover SP-D 35 (both CEM Corp.). This conclusion is clearly reflected by the comparison of |zeta| scores between both microwave systems (Table 3).”

Comment/Question: Also the method of temperature measurement is not given (internal or external). I would like to know why 2 different instruments with different digesting mixtures have been used, what are the limitations of those instruments with respect to the turboWAVE and what would be the best digestion protocol for the CRMs. 

Answer: The major differences (which directly indicate their advantages and disadvantages) between the microwave systems are now described in more detail in the M&M part:

The following section has been modified: L 159 – 174: “The three microwave systems compared in this study differ in the general construction, but the main practical differences refer to the number of vessels that can be processed at a time, the vessel sizes (section 2.2) and the pressure as well as temperature regulation. Briefly summarized, the MARS 6 and the Discover SP-D 35 (external IR temperature control; pressure vessels) used in this study enable digestion at temperatures up to 230 °C and observed pressures up to 24 - 28 bar, whereas the turboWAVE bears a significantly higher maximum temperature of 300 °C and also a significantly higher maximum pressure of 200 bar. In the tuboWAVE, Temperature and pressure are both regulated and controlled in a single reaction chamber filled with inert gas. In contrast to the MARS 6 and the turboWAVE microwave, that feature simultaneous processing of a batch of digestions (40 and 15 vessels), the Discover SP D 35 (in conjunction with an Explorer autosampler) irradiates the vessels automatically one after another enabling variation of digestion parameters for method development (different conditions for every vessel possible). Please note that this comparison is not meant to be exhaustive, since there are a lot of different vessel types (e.g. for different maximum pressures and temperatures), add-ons features (e.g. for pressure and temperature control) and also other microwave systems available on the market.”

Comment/Question: Can a general protocol be applied, or does the protocol need to be adapted depending on the CRM and type of MW instrument, respectively? A bit more detailled information on such a protocol would be desirable, also with respect to reproducibility.

Answer: Thank you very much! This aspect was also addressed by the other reviewers. We have now implemented two recommended protocols depending on how obstinate the polymer matrix is. The first protocol is sufficient for the majority of synthetic polymeric materials (L 450 – 452: “Therefore, for studies focusing on electrotechnical waste and other very obstinate plastic matrices, we recommend the use of the 2nd proposed protocol (Table 5).”)

The following section has been modified: L 469 – 472: “For future studies investigating the interactions (sorption and desorption processes and the inherent metal content) between the most common environmental particulate plastic types and metal contaminants, we recommend the use of the 1st proposed MWAD protocol (Table 5) for total acid digestion.”

L 475– 477: Table 5

Minor comments:

Comment/Question: 1) Which cleaning procedure was applied for the MW vials?

Answer: L209 – 211: “…Microwave vessels were cleaned by running the respective MWAD program two times solely with 4 mL HNO3 and 1 mL HCl (without CRM). Subsequently, the vessels were washed 3-times with Milli-Q water.”

Comment/Question: 2) page 8, line 169: specify TFM

Answer: The following section has been modified: L 224 -225: “…55 mL MARSXpress TFM® (trade name; cross-linked [(CF2)4-CF(-O-CF2-CF2-CF3)-(CF2)5]n) bombs…”

Comment/Question: 3) page 8, line 170: I would suggest a rewording of "CRMs were submerged with ..." to "The respective amount of ... was added to the MW vessels containg the CRMs"

Answer: The following section has been modified L 226 – 227: “The respective amount of HNO3, HCl, H2O2 and HBF4 (section 2.1) was added to the microwave vessels containing the CRM.”

Comment/Question: 4) page 11, line 240: it should be 4-nitrobenzoic acid

Answer: Adapted

Comment/Question: 5) page 15, line 276: ...be explained by the presence of the metal species... I would rather say: ...be explained by the nature of the metal species...

Answer: Adapted

Comment/Question: 6) page 19, line 327: I would recommend an ascending labelling of the SI Tables according to the text. Tables A5+A6 appear last in the text but not in the SI

Answer: Thank you for the thorough reading! Adapted.

Comment/Question: 7) page 19, Figure 4: make the pie chart clearer for Pb in BAM-H010, one does not clearly see to which legend the 0.04% and 0.6% belong

Answer: Adapted

Comment/Question: 8) Figure A1 in the SI: I do not see the acidic proton at 13.30 ppm in the 1H NMR spectrum!!! I also don´t think that according only to that NMR, the structure of 4-nitrobenzoic acid can be confirmed.

Answer: The signal of the acidic proton is strongly broadened due to proton exchange which is fast on the NMR timescale (thus not integrable). The related section of the spectrum including the acidic proton was added and is now enlarged shown in the figure.

Also, the obtained NMR spectrum of the substance was compared to a spectrum of commercially available 4-nitrobenzoic acid (Merck) and found to be similar. The reference spectrum was added into the figure (blue spectrum, SI), too, to confirm the structure.

---

## [Editor Report · Decision Letter 1]

30 Jun 2020

A metrologically traceable protocol for the quantification of trace metals in different types of microplastic

PONE-D-20-09074R1

Dear Dr. Hildebrandt,

We’re pleased to inform you that your manuscript has been judged scientifically suitable for publication and will be formally accepted for publication once it meets all outstanding technical requirements.

Kind regards,

Martin Koller, Ph.D.

Academic Editor

PLOS ONE

Additional Editor Comments (optional):

The comments of all three independent referees were addressed by the authors in a satisfying manner.
---

## [Editor Report · Acceptance letter]

6 Jul 2020

PONE-D-20-09074R1 

A metrologically traceable protocol for the quantification of trace metals in different types of microplastic 

Dear Dr. Hildebrandt:

I'm pleased to inform you that your manuscript has been deemed suitable for publication in PLOS ONE. Congratulations! Your manuscript is now with our production department. 

Kind regards, 

on behalf of

Dr. Martin Koller 

Academic Editor

PLOS ONE